# Valproic Acid-Induced CCN1 Promotes Osteogenic Differentiation by Increasing CCN1 Protein Stability through HDAC1 Inhibition in Tonsil-Derived Mesenchymal Stem Cells

**DOI:** 10.3390/cells11030534

**Published:** 2022-02-03

**Authors:** Yeonsil Yu, Se-Young Oh, Ha Yeong Kim, Ji-Young Choi, Sangmee Ahn Jo, Inho Jo

**Affiliations:** 1Department of Molecular Medicine, College of Medicine, Ewha Womans University, 25 Magokdong-ro-2-gil, Gangseo-gu, Seoul 07804, Korea; yuys@kangstem.com (Y.Y.); ohs@ewha.ac.kr (S.-Y.O.); 2Global Business Development Division, Kangstem Biotech Co., Ltd., 512 Teheran-ro, Gangnam-gu, Seoul 06179, Korea; 3Graduate Program in System Health Science and Engineering, Ewha Womans University, 25 Magokdong-ro-2-gil, Gangseo-gu, Seoul 07804, Korea; 4Department of Otorhinolaryngology-Head and Neck Surgery, College of Medicine, Ewha Womans University, 1071 Anyangcheon-ro, Yangcheon-gu, Seoul 07985, Korea; gkgkgk9402@hanmail.net; 5Department of Nanobiomedical Science & BK21 FOUR NBM Global Research Center for Regenerative Medicine, Dankook University, Cheonan 31116, Korea; jiyoung0220@gmail.com; 6Department of Pharmacology, College of Pharmacy, Dankook University, Cheonan 31116, Korea

**Keywords:** tonsil-derived mesenchymal stem cells (TMSCs), valproic acid (VPA), CCN1, osteogenesis

## Abstract

Our previous study found that the level of CCN1 increases as osteogenic differentiation progresses in tonsil-derived mesenchymal stem cells (TMSCs). This study investigated how CCN1 is regulated through HDAC inhibition in TMSCs and their relationship with osteogenesis. Valproic acid (VPA) (1–5 mM), a well-known histone deacetylase (HDAC) inhibitor, strongly inhibited TMSC proliferation without altering MSC-specific surface markers, CD14, 34, 45, 73, 90 and 105. However, CD146 expression increased at 5 mM VPA. VPA increased osteogenic differentiation of TMSCs but decreased adipogenesis and chondrogenesis, as evidenced by the cell-specific staining of differentiation. The former was validated by the increased osteocalcin (OCN). The changes in CCN1 by VPA was biphasic; it increased until 48 h and decreased thereafter. Knockdown of CCN1 by using siRNA inhibited the osteogenic effect of VPA. VPA had no effect on *CCN1* mRNA expression, but inhibition of protein synthesis by cycloheximide showed that VPA slowed down the CCN1 protein degradation. Moreover, overexpression of HDAC1 completely inhibited VPA-induced CCN1. Our results indicate that VPA inhibits the HDAC1, inducing CCN1 protein stability rather than gene expression, thereby promoting osteogenic differentiation of TMSCs. These findings present the noble implication of VPA as an inhibitor of HDAC1 to facilitate CCN1-induced osteogenic differentiation of MSCs.

## 1. Introduction

Tonsil-derived mesenchymal stem cells (TMSCs) are multipotent MSCs that have increasingly been studied due to their relatively high proliferation rate and low allogenicity compared to other tissue-derived MSCs [1]. Previous studies determined that TMSCs have differentiation potential into mesodermal lineage, as well as endodermal and ectodermal lineages, placing TMSCs as a new promising therapeutic tool for regenerative medicine [1,2,3,4]. To increase the efficacy of the TMSCs for future applications, we have been studying functional characteristics of the TMSCs and found that the CCN1 inhibition by siRNA significantly reduced the osteogenesis of the TMSCs [5], pointing out the importance of CCN1 in osteogenic differentiation.

The acronym “CCN” was termed after the first three members of the family: Cyr61 (cysteine-rich protein 61 or CCN1), CTGF (connective tissue growth factor or CCN2), and NOV (nephroblastoma overexpressed gene or CCN3) [6]. The CCN protein family, including CCN1, is a group of multifunctional regulatory proteins that are involved in various biological activities, including proliferation, senescence, adhesion, migration and cell differentiation [7,8]. In resemblance to the results of our study, a few numbers of studies have also been reporting the potential contribution of CCN1 in cell differentiation. For example, CCN1 was shown to promote the differentiation of endothelial progenitor cells [7], and a more recent study reported that CCN1 stimulated the proliferation and differentiation of osteoblasts, altering bone remodeling of myeloma bone disease [8]. The CCN1 could be regulated in many different ways, and it seems that various stress conditions, such as hypoxia [9], UV light [10] and mechanical strain [11], are the factors that could induce the CCN1 expression. Mechanical strain, for example, induces acetylation of histone 3 and 4 at the CCN1 gene-coding region by activating CREB-binding protein (CBP) histone acetyltransferase (HAT) through signaling cascade of p38 stress-activated protein kinase (SAPK), thereby increasing promoter activity and expression of CCN1 [11]. CCN1 has been shown to trigger cellular senescence in fibroblasts, while promoting the regeneration of other tissues [12]. Additionally, the results from our study indicated that the expression of CCN1 is required for the proper induction for the osteogenic differentiation of TMSCs [7], which made us to further investigate the detailed molecular mechanism associated with CCN1 and osteogenesis.

As described earlier, the acetylation of histone plays an important role in regulating CCN1 expression, and, therefore, we speculated that inhibition of histone deacetylases (HDACs), which are known to have the opposite function of CBP HAT on acetylation, could upregulate the CCN1 activity/expression, inducing osteogenesis. Koon et al. (2011) also showed the possible link between CCN1 and HDAC activities [13]. To investigate these effects and their association, valproic acid (VPA) was selected as an inhibitor of HDACs. VPA has already clinically been used as an anticonvulsant and mood-stabilizing drug for treating epilepsy and bipolar disorder [14]. Recent evidence also showed that HDAC inhibitors are used as a potential treatment for HIV and cancer by reversing the HIV latency in cells and proliferation [15,16]. HDACs are also known to be important regulators of cell differentiation. Few studies have demonstrated that VPA inhibits the HDAC activities, subsequently increasing osteogenic and hepatogenic differentiation but decreasing adipogenic, chondrogenic and neurogenic differentiation [17,18,19]. Nevertheless, the mechanism involved in cell differentiation has not yet been identified. Therefore, this study sought to investigate the potential effects of VPA on CCN1 as an HDAC inhibitor and to identify the roles of HDAC and CCN1 in osteogenesis, using TMSCs.

## 2. Materials and Methods

### 2.1. TMSC Isolation and Culture

TMSCs were isolated and cultured as described, with minor modifications [2]. In brief, tonsil tissues were obtained from patients (≤10 years) undergoing tonsillectomy, and informed written consent was obtained from all patients and/or their legal representatives. The study protocol was approved by the Ewha Womans University Medical Center institutional review board (ECT-11-53-02), the date of approval is 22 September 2011.

The tonsillar tissue was minced and digested in Roswell Park Memorial Institute medium 1640 (RPMI-1640; Invitrogen, Carlsbad, CA, USA) containing 210 U/mL collagenase type I (Invitrogen) and 10 g/mL DNase (Sigma-Aldrich, St Louis, MO, USA) for 30 min at 37 °C on a magnetic stirrer. 

The digested tissues were filtered through a wire mesh with a pore size of 20 μm. The cells were then washed twice in Dulbecco’s-modified Eagle’s medium-high glucose (DMEM-HG; Thermo Fisher Scientific, Waltham, MA, USA) supplemented with 20% fetal bovine serum (FBS; Welgene, Daegu, Korea) and once by DMEM-HGs supplemented with 10% FBS. The cells were suspended in phosphate-buffered saline (PBS) and were over-laid on Ficoll-Paque^TM^ PLUS (density 1.077 g/mL; GE Healthcare, Little Chalfont, UK) to obtain mononuclear cells. After the density gradient centrifugation of the digested tonsil tissue at 500 g forces, the mononuclear cells were washed with PBS and resuspended in DMEM-HG supplemented with 10% FBS, as well as 100 U/mL of streptomycin and penicillin and incubated in T-150 culture flasks (Corning Incorporated, Corning, NY, USA) at 37 °C in a 5% CO_2_ humidified incubator. The TMSCs were initially cultured for 2 weeks to remove the dendritic cell population, and the cells were detached with 0.05% of trypsin for 5 min at the maximum to prevent contamination of macrophages and other adherent cells. After 2 weeks of culturing and passaging of the cells, the adherent mononuclear cells, which are designated as TMSCs, hereafter, were collected and maintained in tissue culture dishes in DMEM with low glucose (DMEM-LG; Thermo Fisher Scientific, Waltham, MA, USA), with 10% FBS. 

### 2.2. MTT Cell-Proliferation Assay 

TMSCs (5.0 × 10^3^ cells) were plated in 96-well plates and exposed to various concentrations (0–5 mM) of VPA for indicated times (0–72 h). After the VPA exposure, MTT (0.5 mg/mL) assay solution was added and incubated for 3 h. The solution was removed, formazan crystals were dissolved in DMSO and the absorbance was measured at 540 nm, using Synergy H1M (Biotek, Winooski, VT, USA).

### 2.3. Bromodeoxyuridine (BrdU) Incorporation

TMSCs (5.0 × 10^3^ cells) were plated in 96-well plates and exposed to various concentrations (0–5 mM) of VPA for indicated times (0–72 h). After the VPA exposure, the TMSCs were incubated in BrdU solution for 3 h. After removing the solution, a fixing/denaturing solution was added and incubated for 30 min. The solution was removed, and the cells were incubated in BrdU detection antibody for 1 h. The cells were washed with wash buffer 3 times. The HRP-conjugated secondary antibody solution was added and incubated for 30 min. After washing 3 times with PBS, TMB substrate, followed by stop solution, was added for colorimetric assay, and the absorbance was measured at 450 nm, using Synergy H1M (Biotek).

### 2.4. Apoptosis Analysis

To examine the apoptotic effect of VPA on the TMSCs, TMSCs (1.0 × 10^5^ cells) were exposed to various concentrations (0–5 mM) of VPA for indicated times (0–72 h). The cells were washed twice with cold PBS and resuspended in binding buffer (0.01 M HEPES, 0.14 M NaCl, 0.25 mM CaCl_2_). The cells were incubated with annexin V-FITC (BD Pharmingen Inc, San Diego, CA, USA) for 15 min at room temperature in the dark, followed by labeling with 7AAD (BD Pharmingen) and then analyzed by using flow cytometry analyses.

### 2.5. Flow Cytometry Analysis

To analyze the surface immunophenotypes of the TMSCs, approximately 5.0 × 10^4^ cells (in 100 μL PBS/0.5% bovine serum albumin (BSA)/0.01% NaN_3_) were incubated for 30 min at 4 °C with fluorescein isothiocyanate (FITC)-labeled antibodies against human CD34, CD45, CD90, CD105, CD31 (BD Biosciences, CA, USA), phycoerythrin (PE)-labeled antibodies against human CD14, CD73, CD146 (BD Biosciences) or the respective isotype controls (BD Biosciences). After washing the cells, the labeled cells were analyzed by using the FACScalibur flow cytometer (BD Biosciences). 

### 2.6. Adipogenic, Chondrogenic and Osteogenic Differentiation

The TMSCs were seeded at the density of 1.0 × 10^4^ cells/cm^2^ in 6- and 24-well plates and exposed to 0 or 2.5 mM of VPA for 72 h. The cells were then washed with PBS three times and incubated in the commercially available adipogenic, chondrogenic and osteogenic differentiation media (Thermo Fisher Scientific, Waltham, MA, USA). After 2 weeks of the differentiation, the cells were fixed with 4% PFA for 1 h and stained with their respective detection solution as follows. The adipogenicity was visualized by staining intracellular lipid droplet formation with 2% Oil Red O (Sigma-Aldrich) for 1 h at room temperature. The chondrogenicity was visualized by staining sulfated proteoglycan deposits with 1% Alcian blue (Sigma-Aldrich) for 1 h at room temperature. The osteogenicity was visualized by staining calcium mineral deposits with 2% Alizarin Red S (Sigma-Aldrich) for 1 h at room temperature. 

### 2.7. Reverse-Transcription Polymerase Chain Reaction (RT-PCR) Assay 

Total RNA was extracted from TMSCs, using TRIzol^®^ RNA Isolation reagents (Thermo Fisher Scientific, Waltham, MA, USA), and 0.2 μg of RNA was reversely transcribed by using oligomers. PCR was performed by using Platinum PCR SuperMix (Thermo Fisher Scientific, Waltham, MA, USA). Gene-specific primers (forward/reverse) were designed based on cDNA sequences of GenBank. Primers used in the experiment were as follows: *CCN1,* 5’-CAACCCTTTACAAGGCCAGA -3’ (forward) and 5’-TGTAGAAGGGAAACGCTGCT-3’ (reverse); *GAPDH*, 5’-GGAGCGAGATCCCTCCAAAAT-3’ (forward) and 5’- GGCTGTTGTCATACTTCTCATGG-3’ (reverse).

### 2.8. Western Blot Analysis

Whole protein lysates were extracted by using lysis buffer (20 mM Tris pH 7.5, 150 mM NaCl, 1 mM EDTA, 1 mM EGTA, 1% Triton-X 100, 0.1 mM Na_3_VO_4_, 2 μg/mL leupeptin and 100 μg /mL PMSF). Protein concentrations were measured by a BCA protein assay (Sigma-Aldrich). Equal quantities of protein (20 μg) were separated on 8–15% SDS–PAGE after the transfer of the protein onto the nitrocellulose membrane. The blots were then probed with a primary antibody against osteocalcin (OCN), osteopontin (OPN), RUNX2, CCN1 and PDZ-binding motif (TAZ), followed by their corresponding secondary antibodies. The protein blots were developed by using enhanced chemiluminescence reagents (Amersham, Buckinghamshire, UK). The protein blots were quantified by using Image Lab software (Bio-Rad, Hercules, CA, USA).

To further evaluate the effect of VPA on the CCN1 protein stability, the CCN1 protein synthesis was inhibited by treating the TMSCs with 20 μM of cycloheximide after 24 h of either 0 or 2.5 mM of VPA exposure. The time-dependent degradation of CCN1 was evaluated at 0–30 min after the cycloheximide treatment. The protein level of CCN1 was quantified by using Image Lab software and normalized by the protein level of GAPDH. The values represent mean ± SD of the fold changes relative to the control at 0 min of time points for each respective 0 and 2.5 mM of VPA treatment.

### 2.9. CCN1 Enzyme-Linked Immunosorbent Assay (ELISA)

TMSCs were exposed to various concentrations (0–5 mM) of VPA for indicated times (0–72 h). The concentrations of CCN1 in the culture supernatant were measured by ELISA according to the manufacturer’s instruction (R&D Systems, Minneapolis, MN, USA). Briefly, the culture supernatants of VPA-treated TMSCs were collected and added onto antibody-coated wells. After 2 h of incubation at room temperature, the wells were washed four times with wash buffer; then human CCN1 conjugates were added and incubated overnight. The wells were washed four times with wash buffer, followed by substrate buffer, and incubation was continued for 30 more minutes at room temperature. The relative absorbance was measured at 450 nm, using Synergy H1M (Biotek, Winooski, VT, USA). The CCN1 concentrations were extrapolated from a standard curve.

### 2.10. Transfection

For a stable transfection, the TMSCs were grown to reach 80% confluency in 6- or 24-well plates. Before the transfection, the TMSCs were washed with PBS three times and were incubated in serum-free Opti -MEM (Thermo Fisher Scientific, Waltham, MA, USA) containing either 100 nM control or CCN1 siRNA (Catalogue No. SC-37007, Santa Cruz Biotechnology, Dallas, TX, USA) and DharmaFECT1 reagent for 24 h. The transfection using DharmaFECT1 was performed according to the protocol provided in the supplier’s manual (Fisher Scientific, Nazareth, PA, USA). After the transfection, the TMSCs were treated with 0 or 2.5 mM of VPA for 24 h and further cultured in osteogenic medium (Thermo Fisher Scientific, Waltham, MA, USA) for 1 or 2 weeks to assess the osteogenic differentiation. The level of osteogenesis was assessed by comparing OCN protein level and calcium mineralization determined by Western blot and Alizarin Red S staining, respectively.

For HDAC1 and HDAC2 overexpression, pOTB7 vector including HDAC1 and pME18s-FL3 vector including HDAC2 were purchased from Korea Human Gene Bank (Korean BioInformation Center, Daejeon, Korea). TMSCs (1.0 × 10^6^ cells) were transfected with either 2 μg of HDAC1/pOTB7 or 2 μg of HDAC2/pME18S-FL3 by electroporation, using an Amaxa Human MSC Nucleofector^®^ kit (Lonza, Portsmouth, NH, USA) with Amaxa Scientific Nucleofector^TM^ II Device (Lonza, Portsmouth, NH, USA), according to the manufacturer’s protocol. After the transfection, the CCN1 protein was determined by Western blot analyses. 

### 2.11. Statistical Analysis 

All data are presented as mean ± standard deviation (SD). Statistical significance among different concentrations was determined by one-way ANOVA, followed by TUKEY’S analysis, using GraphPad Prism 7.0 software (GraphPad Software Inc. La Jolla, CA, USA). The differences between the two groups were compared by Student’s *t*-test. Statistically significant values are denoted as * *p* < 0.05, ** *p* < 0.01 or *** *p* < 0.001.

## 3. Results

### 3.1. VPA Decreases the Proliferation of TMSCs by Inhibiting DNA Synthesis

The doubling time of the TMSCs was estimated to be 38 h [5,20], and, therefore, 72 h of exposure period was chosen to evaluate the effects of VPA on the proliferation of TMSCs. MTT assay showed that VPA exposure for 72 h decreased the proliferation of the TMSCs in a concentration-dependent manner, where its effect was significant at 1 mM (*p* < 0.05), 2.5 mM (*p* < 0.01) and 5 mM (*p* < 0.001) (Figure 1A). However, the proliferation was slightly increased at 100 μM (*p* < 0.01) but had no significant effect at concentrations less than 10 μM (Appendix A). To further evaluate the cytostatic effects of VPA, the TMSCs were stained with exogenous BrdU assay solution to look at the effect of VPA on DNA synthesis. Then 1, 2.5 and 5 mM of VPA exposure for 72 h decreased the level of exogenous BrdU in the same way as the proliferation, suggesting that VPA decreases the proliferation of the TMSCs by inhibiting the DNA synthesis (Figure 1B).

The effect of VPA on the viability of the TMSCs was also assessed by performing flow cytometry analyses on the TMSCs stained with annexin-V and 7-AAD. The percent of the live TMSCs after 2.5 mM VPA exposure for 72 h was 93 ± 0.68% (double-stained with AV and 7-AAD), which was not significantly different compared to those cells exposed to 1 mM of VPA (Figure 1C). However, 72 h of VPA exposure at 5 mM further induced the cytotoxicity of VPA, leading to the viability of the TMSCs less than 90% (Figure 1C). The morphological changes mediated by the VPA were observed under a light microscope (40×). As expected from the previous MTT assays, the TMSCs exposed to VPA proliferated slowly and had stretched and flattened shapes at 48 and 72 h of post-exposure (Figure 1D).

### 3.2. Immunophenotypic Profiling of the TMSCs Exposed to VPA

According to the minimal standard criteria to define human MSCs set by the International Society for Cellular Therapy (ISCT) [21], MSCs must have the positive surface markers, CD73/5’-Nucleotidase, CD90/Thy1, CD105/Endoglin and CD44/homing cell adhesion molecule (HCAM), but in the absence of the following hematopoietic lineage markers: CD14, CD34, CD45, CD11b/Integrin alpha M, CD19, CD79a and HLA Class II, as well as endothelial markers, CD31 and CD146. Therefore, flow cytometry was performed to determine whether VPA could influence the MSC markers of the TMSCs.

As shown in Figure 2, the control TMSCs without VPA exposure had a high expression of MSC positive markers including CD44, CD73, CD90 and CD105, but had a low expression of negative markers, CD14, CD31, CD34, CD45 and CD146. A total of 2.5 mM of VPA exposure for 72 h had no significant effect on most of MSC surface expression, but significantly increased the expression of CD146 on the TMSCs from 2.80 ± 0.54% to 15.61 ± 0.18% (*p* < 0.05) (Figure 2). We further tested other concentrations of VPA to look at its effect on CD146 expression of the TMSCs. VPA at 1 or 100 μM did not affect CD146 expression compared to the control TMSCs. On the contrary, VPA at 2.5 and 5 mM dramatically increased from 13.53% to 44.99% and 62.39%, respectively, compared to the control TMSCs (Appendix A).

### 3.3. VPA Increases Osteogenic Differentiation but Decreases Adipogenic and Chondrogenic Differentiation of the TMSCs

To examine the effect of VPA on the differentiation potentials of the TMSCs, the TMSCs were exposed to 2.5 mM of VPA for 72 h and subsequently incubated in commercially available adipogenic, chondrogenic and osteogenic media for 2 weeks. Our results showed that VPA increased the osteogenic differentiation of the TMSCs as visualized by the calcium mineralization stained by Alizarin Red S (Figure 3A), even though it decreased the adipogenic and chondrogenic differentiation of the TMSCs, and this was assessed by the decreased formation of lipid droplets stained with Oil Red O and sulfated proteoglycan deposits stained with Alcian blue, respectively (Figure 3A). Western blotting analyses also validated the osteogenic inducibility of VPA, where the protein level of osteocalcin (OCN), an osteogenic marker, was significantly increased at 4 and 14 days of the osteogenic differentiation (Figure 3B).

### 3.4. VPA Induces the Protein Level of CCN1 in a Dose-Dependent Manner but Not the Protein Level of TAZ, OPN and RUNX2

To determine the mechanism of action of VPA in inducing osteogenesis of the TMSCs, some osteogenic inducing factors were assessed. The TMSCs were exposed at 1, 2.5 and 5 mM of VPA for 72 h, and the protein level of CCN1 [5], TAZ [22], OPN [23] and RUNX2 [24] was analyzed by using Western blotting. VPA slightly decreased the protein level of TAZ, RUNX2 and OPN in a dose-dependent manner, but its effect was not significant (Figure 4). However, VPA significantly increased CCN1, a known angiogenic and osteogenic factor, at 2.5 and 5 mM VPA exposure (*p* < 0.001), suggesting that VPA could induce osteogenesis by inducing the CCN1 expression. 

### 3.5. VPA Induces the Osteogenesis of the TMSCs by Inducing the CCN1 Level

To investigate the potential link between the CCN1 and the osteogenesis induced by VPA, the TMSCs were transfected with either siRNA against control gene (siControl) or siCCN1 during 72 h of 2.5 mM VPA exposure, followed by subsequent incubation in commercially available osteogenic media for 1 or 2 weeks. As shown in Figure 5A, siCCN1 successfully reduced the protein level of CCN1, and the downregulation of CCN1 decreased the calcium mineralization (Figure 5B), as well as the OCN protein level compared to the VPA-treated TMSCs transfected with siControl (Figure 5C). These results indicate that VPA induces the osteogenic differentiation of the TMSCs by increasing the CCN1 protein level.

### 3.6. VPA Increases the Protein Level of CCN1 by Inducing the Stability of the CCN1 Protein

Considering that CCN1 is one of the important factors of osteogenesis induced by VPA, we investigated the mechanism by which VPA increases CCN1 expression. We initially measured the media concentration of CCN1 released from the TMSCs seeded at the density of 2.0 × 10^5^ cells/mL followed by VPA exposure at 0, 1, 2.5 and 5 mM for 24, 48 and 72 h. 

The CCN1 concentration in the media significantly increased even at 24 h of VPA exposure, where its effect was the highest at 1.0 mM (*p* < 0.001), followed by 5.0 (*p* < 0.001) and 2.5 mM (*p* < 0.01) (Figure 6A). The media concentration of CCN1 was more dramatically increased when the TMSCs were exposed to VPA for 48 h (*p* < 0.001). However, the CCN1 secretion was diminished after 72 h of VPA exposure, where the CCN1 concentration was not significantly different at 1 and 2.5 mM compared to the control but remained to be significant at 5 mM of VPA exposure. A similar pattern was also observed with the protein level of the TMSCs analyzed by using the Western blotting analyses, where the induction of CCN1 was the highest at 48 h of VPA exposure (Figure 6B). 

We performed ChIP assays to determine the potential contribution of histone acetylation at histone 3 (H3) on the VPA-mediated induction of CCN1 and found that VPA induced the binding of AcH3K18 at *CCN1* promoter (Appendix A). It was also determined that VPA did not alter the phosphorylation of ERK, JNK and p38 (Appendix A). Interestingly, VPA did not affect the expression of the *CCN1* transcription (Figure 6C), suggesting that VPA induces the CCN1 expression at the translational level.

To further confirm the effect of VPA on the translation of *CCN1*, the CCN1 protein synthesis was inhibited by treating the TMSCs with 20 μM of cycloheximide after 24 h of either 0 or 2.5 mM of VPA exposure. The time-dependent degradation of CCN1 was evaluated at 0, 10, 20 and 30 min after the cycloheximide treatment, and we found that the degradation rate of CCN1 from the TMSCs treated with 2.5 mM VPA was much slower compared to that of the control TMSCs (Figure 6C), indicating that VPA increased the overall CCN1 protein level by increasing the stability of the CCN1 protein but not by increasing the transcription.

### 3.7. VPA Induces the CCN1 by Inhibiting Histone Deacetylase 1 (HDAC1) but Not HDAC2

It was reported that the acetylation of histones 3 and 4 at the *CCN1* gene-coding region induces the CCN1 expression [10]. Since VPA is a well-known HDAC inhibitor [25], we also looked at the potential involvement of HDACs on CCN1 expression. The TMSCs exposed to 2.5 mM of VPA were transfected with either HDAC1 or HDAC2 overexpression vector, and the protein level of CCN1 was analyzed by using Western blotting analyses. The overexpression of HDAC2 did not affect the level of CCN1 induced by VPA, yet the overexpression of HDAC1 dramatically decreased the CCN1, indicating that HDAC1 is the primary regulatory enzyme of the CCN1 expression (Figure 7).

## 4. Discussion

For the last few years, our research group has been looking at the functional characteristics and phenotypes of the TMSCs, regulating cell proliferation and differentiation. Our previous study determined that the continual passages of TMSCs until passage 10 induced the osteogenesis of the TMSCs, which were closely associated with the CCN1 expression [5]. The knockdown of CCN1 by using siRNA decreased the osteogenic differentiation of the TMSCs, introducing CCN1 as an important regulator of osteogenesis in relation to cellular senescence. 

A few studies showed that the acetylation of histone 3 is crucial for regulating the *CCN1* gene expression [11,13], and, therefore, we selected the valproic acid (VPA), an HDAC inhibitor, to investigate the potential association among CCN1 and osteogenic differentiation of the TMSCs. HDACs can be classified into four types, depending on their expression and function [26,27]. Class I HDACs are ubiquitously expressed throughout the whole-body tissues, and they function as transcriptional regulators. Class II, III and IV HDACs contribute to tissue-specific function, formation of aggresome and autophagy, and cytokine regulation. However, each of these HDACs may have overlapped or their specific function in regulating various cellular activities, altering cell survival, apoptosis and epigenetic regulation [26,27]. The VPA used in this study is known to specifically target class I HDACs, inhibiting the activity of HDAC1, 2 and 3 [28].

Our initial study determined that VPA has a strong cytostatic effect at a concentration higher than 1.0 mM, where VPA inhibited the proliferation of the TMSCs by hindering the DNA replication without altering MSC immunophenotypes. Several studies have already addressed the mechanism of VPA regarding the anti-proliferative effect [29,30,31]. VPA inhibited the proliferation of breast cancer cell line SKBR3 by arresting the cell cycle through the acetylation of Hsp70 [29]. Another study also showed the arrest of the cell cycle in primary murine prostate cancer cells, due to the increased expression of cyclin D2, a known tumor suppressor protein [31]. Apart from these studies, Venkataramani et al. (2010) reported β-amyloid precursor protein (APP) as the target of VPA, where its downregulation through HDAC inhibition decreased the proliferation of human pancreatic adenocarcinoma and colon cancer cells [30]. Although these studies focus on different target molecules to describe the inhibitory effect of VPA on cell proliferation, all of them agree that HDAC inhibition is upstream of all these target molecules. We also found that the induction of CCN1 is primarily associated with the inhibition of the HDAC1; the overexpression of HDAC1 decreased the CCN1 level induced by VPA exposure in the TMSCs.

In addition to the antitumorigenic activity of HDAC inhibitors, many studies have reported their potential contribution to changing cell differentiation. For instance, vorinostat, which is marketed under the name Zolinza^®^ by Merck, is a potent HDAC inhibitor for treating cutaneous T-cell lymphoma but has been shown to cause bone loss during clinical trials of multiple myeloma [32]. However, Xu et al. (2013) further investigated this adverse effect, concluding that vorinostat was rather inducing the osteogenesis and promoted bone formation in a mouse model [33]. Similar to this, we found that VPA also has the potential to induce osteogenesis by inducing CCN1 expression through the HDAC1 inhibition but not HDAC2, because the overexpression of HDAC1 dramatically decreased the CCN1 expression induced by VPA. The effect of VPA on osteogenesis seems to be still controversial, because VPA was shown to hinder the terminal differentiation of human dental pulp stem cells and osteoblasts by inhibiting HDAC2 [25]. However, another study reported that VPA increased the osteogenesis of human bone-marrow-derived MSCs with the induction of osteogenic markers, including Runx-2, osterix, ALP, OPN and OCN [17]. Furthermore, there is growing evidence from both in vitro and in vivo studies, indicating that the inhibition of HDACs, particularly HDAC1, promotes osteogenic differentiation [17,18,34].

The inhibition of HDAC1 induced the osteogenic differentiation of murine osteoblasts [18], and the authors postulated that HDAC1 regulates the transcriptional activity of *Runx2*, altering the differentiation. We observed a similar finding, where the HDAC1 inhibition induced the osteogenic differentiation of the TMSCs, yet the HDAC1 inhibition did not necessarily promote the gene activities of *Runx2*, *TAZ* and *OPN* [35]. Instead, we found that HDAC1 inhibition by VPA increased the CCN1 protein, which, in turn, induced the osteogenic differentiation of the TMSCs, as evidenced by the increased calcium mineralization and OCN from the TMSCs treated with VPA or CCN1 siRNA. Since OPN is also known as a marker at the terminal stage of osteogenic differentiation, it may not be necessarily induced at a very early stage of the osteogenic differentiation [36]. Nevertheless, there is no plausible explanation of how the induction of CCN1 by VPA increased the osteogenesis without the elevation of Runx2 and TAZ the well-known pre-osteogenic factors. Matsubara et al. (2008) reported the increased osteogenic differentiation of the Runx2-deficient MSCs through osterix (Osx) and Msx2 [37], pointing out the Runx2-independent regulation of the osteogenesis in MSCs. Therefore, it can be speculated that the CCN1 directly induces another pre-osteogenic factor known as Osx, independent of Runx2 [37,38], even though further studies need to be performed to find the potential association between CCN1 and Osx.

It is interesting to note that VPA does not necessarily elicit the stress response, similar to other stress-stimulatory factors. For example, a mechanical strain was shown to induce the acetylation of histone 3 at the CCN1 gene-coding region by activating CBP HAT through the cascade of p38 stress-activated protein kinase (SAPK), subsequently increasing promoter activity and expression of CCN1, as shown in Appendix A [11]. However, VPA did not induce the phosphorylation of ERK, JNK and p38, regardless of the increased binding activity and protein level of CCN1 (Appendix A), suggesting that the increased CCN1 protein level is independent of ERK and stress-activated JNK-p38 pathway. This is probably expected, considering that the mRNA expression of CCN1 was not necessarily induced by VPA, indicating that VPA induces CCN1 protein in another way. Substance P was shown to induce CCN1 expression via activating mitogen-activated protein kinase (MAPK) [39], and, therefore, VPA exposure could have stimulated other pathways that are independent of the ERK–JNK–p38 pathway. 

The most plausible regulatory mechanism in relation to CCN1 would be the autophagic degradation of CCN1 mediated by p62-dependent pathway. Pterostilbene, a plant-extracted antioxidant in the implication of anti-carcinogenesis, was shown to have the exact opposite pattern of VPA on CCN1, where the extract inhibited the ethanol-induced CCN1 through post-transcriptional regulation without altering the gene transcription of *CCN1*, preventing cellular senescence induced by ethanol in murine liver [40]. Since the level of CCN1 declined after the 15th passage, it would be difficult to conclude CCN1 as a strong senescent marker [5]. Nevertheless, we found that both replicative senescence and VPA induced the CCN1 by increasing protein stability rather than regulating the gene expression. Therefore, it could be speculated that VPA inhibits p62-dependent autophagy in opposition to pterostilbene. Wu and his colleagues (2021) demonstrated that the mTOR pathway is activated by cholesterol derivatives that phosphorylate HDAC1 [41]. HDAC1 deacetylates autophagy-related protein 4b (Atg4b), which catalyzes the ubiquitin-like system, LC3/Atg8–phosphatidylethanolamine (PE), to generate autophagosome for degrading various cellular components [42]. Based on the findings of these studies altogether, VPA is likely to inhibit the mTOR pathway, dephosphorylating the HDAC1 [41]. This would prevent the translocation of HDAC1 into the nucleus, increasing acetylation of Atg4b, which is unlikely to catalyze the LC3/Atg8-PE required for autophagosome maturation [42]. The decreased autophagosome would eventually increase the stability of CCN1 protein, increasing the overall activity of CCN1 to promote osteogenesis. The detailed mechanism of CCN1 regarding osteogenesis could not be elucidated in this study. However, previous studies with CCN1 proposed that CCN1 interacts with the integrin α_v_β_3_ to decrease the sclerostin (Sost) expression, promoting the Wnt signaling pathway to induce the osteogenic differentiation of the TMSCs in this study [8,43]. In fact, it has been suggested that the Wnt signaling induces the Osx, which could promote the osteogenic differentiation of the MSCs independent of the Runx2 and TAZ [38]. Nonetheless, these are speculations based on previous findings, and it would be interesting to validate these mechanisms of VPA in CCN1 regulation and osteogenesis.

In addition to the possible interaction between CCN1 and the integrin α_v_β_3_, the increased level of CD146 at high concentrations of VPA should also be noted. Although the overexpression of CD146 was reported to promote cancer progression [44], it also has the function of defining the subpopulation capable of bone formation in BM-MSCs [45] and triggering β1-integrin activation in T cells [46]. Therefore, the induction of CD146 by VPA could further enhance the Wnt signaling pathway through integrin α_v_β_3_ to take part in inducing osteogenic differentiation of the TMSCs.

It has been almost 60 years since the first medical use of VPA for epilepsy therapy, yet there are still many things to uncover about its precise biological role. In this study, we investigated the effect of VPA as an HDAC1 inhibitor in modulating CCN1 expression and found their potential link on osteogenic differentiation of the TMSCs. The results of our study indicate that VPA induces the CCN1 expression through the HDAC1 inhibition, thereby promoting osteogenic differentiation of the TMSCs. Although the detailed mode of action has not yet been elucidated, we discovered the noble implication of VPA as an HDAC1 inhibitor to facilitate CCN1-induced osteogenic differentiation of MSCs in addition to its original pharmacological roles as a drug for treating epilepsy and bipolar disorder [14], as well as a potential anticancer reagent [47]. Since CCN1 is also known to improve the angiogenesis and migration of human umbilical vein endothelial cells through the upregulation of integrin α_v_β_3_ and AMPK [48], there is a possible implication that VPA can improve cardiovascular health.

## Figures and Tables

**Figure 1 cells-11-00534-f001:**
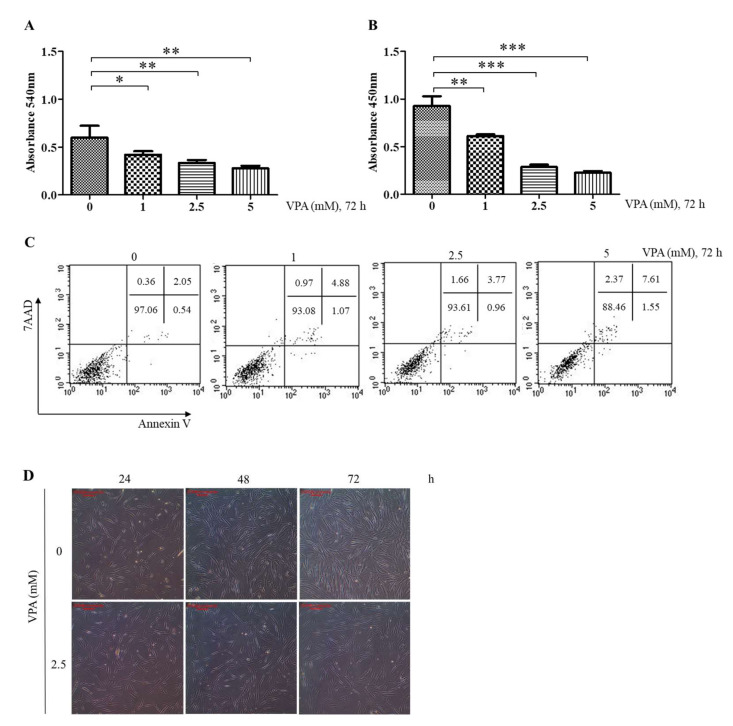
Effect of VPA on the proliferation, cell viability and morphologies of the TMSCs. The Table 0. 2.5 and 5 mM of VPA for 72 h, and the following assays were performed to assess the cytotoxicity of VPA. (**A**) MTT assay to determine the effect of VPA on the proliferation of TMSCs. (**B**) BrdU assay to assess the effect of VPA on the DNA synthesis of the TMSCs. (**C**) Flow cytometry analyses of the TMSCs stained with annexin-V (AV) and 7-aminoactinomycin D (7AAD) to assess the apoptotic effect of VPA on the TMSCs. The values represent mean ± SD of absorbance from three independent experimental trials. Statistically significant are denoted as * *p* < 0.05, ** *p* < 0.01 or *** *p* < 0.001, which were determined by one-way ANOVA, followed by Tukey’s multiple comparisons. (**D**) Morphologies of the TMSCs exposed to VPA at 2.5 mM for 24, 48 and 72 h were observed under a light microscope (40×; scale bar, 200 μm). The results are representative of three independent experimental trials.

**Figure 2 cells-11-00534-f002:**
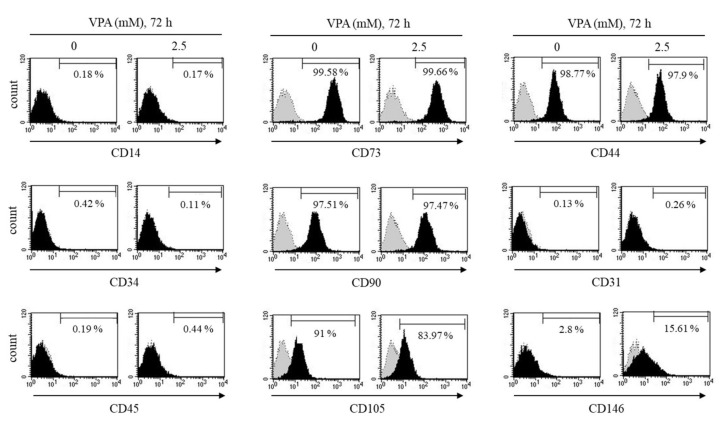
Effect of VPA on the MSC surface markers of the TMSCs. After the TMSCs were exposed to either 0 or 2.5 mM of VPA for 72 h, the cells were tagged with their respective antibodies labeled with either FITC or PE followed by flow cytometry to analyze the positive MSC markers CD73, CD90 and CD105 and the negative MSC markers CD14, CD31, CD34, CD45 and CD146. The results are representative of three independent experimental trials.

**Figure 3 cells-11-00534-f003:**
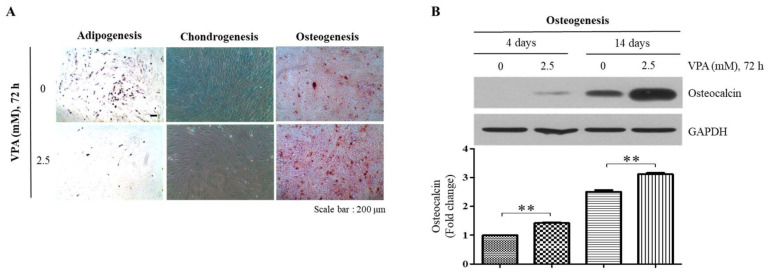
Effect of VPA on the adipogenic, chondrogenic and osteogenic differentiation potentials of the TMSCs. After treating the TMSCs to either 0 or 2.5 mM of VPA for 72 h, the cells were subsequently incubated in the commercially available adipogenic, chondrogenic and osteogenic differentiation media. (**A**) After 2 weeks of the differentiation, the cells were fixed with 4% PFA and stained with 2% Oil Red O, 1% Alcian blue and 2% Alizarin Red S staining solution to assess the adipogenicity, chondrogenicity and osteogenicity of the TMSCs. (**B**) After 4 and 14 days of the osteogenic differentiation, the protein lysates of the TMSCs were collected, and Western blotting was performed to assess the protein level of OCN. The protein level of OCN from the blot was quantified by using Image Lab software and normalized by the level of GAPDH. The values represent mean ± SD of the fold changes relative to the control at 4 days of differentiation from three independent experimental trials. Statistically significant is denoted as ** *p* < 0.01, as determined by using Student’s *t*-test.

**Figure 4 cells-11-00534-f004:**
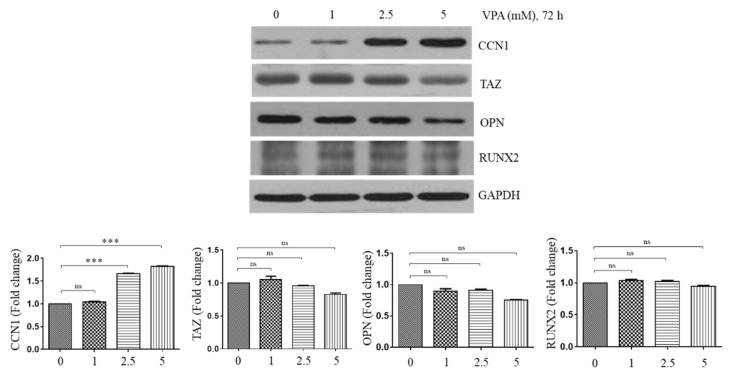
Effect of VPA on the protein level of CCN1, TAZ, OPN and Runx2. The TMSCs were exposed to 0, 1, 2.5 and 5 mM of VPA for 72 h, and the protein lysates of the TMSCs were collected to assess the protein level of the CCN1, TAZ, OPN and Runx2, using Western blotting analyses. The protein level of the blots was quantified by using Image Lab software and was normalized by the protein level of GAPDH. The results are representative of three independent experimental trials, and the values represent mean ± SD of the fold changes relative to the control without VPA exposure. Statistically significant is denoted as *** *p* < 0.001, which were determined by one-way ANOVA followed by Tukey’s multiple comparisons.

**Figure 5 cells-11-00534-f005:**
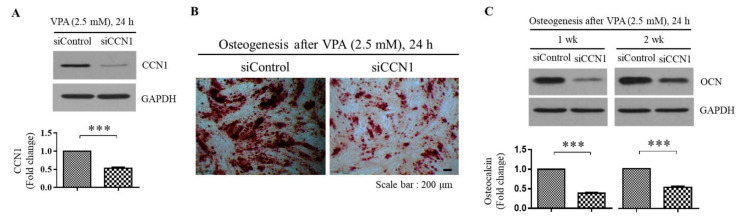
Inhibition of CCN1 by siRNA prevents the osteogenesis induced by VPA exposure. The TMSCs were grown to reach 80% confluency and transfected with either control or CCN1 siRNA, using DharmaFECT1 reagent for 24 h in serum-free Opti-MEM. After the transfection, the TMSCs were treated with 2.5 mM of VPA for 24 h. (**A**) Western blotting was performed to confirm the efficacy of CCN1 siRNA in downregulating the siCCN1 protein level. After the control or CCN1 siRNA transfection, followed by 2.5 mM of VPA treatment, the TMSCs were further cultured in osteogenic medium for 1 or 2 weeks to assess the osteogenic differentiation. The level of osteogenesis was assessed by comparing (**B**) calcium mineralization Alizarin Red S staining and (**C**) OCN protein level by Western blotting analyses. For the Western blotting analyses, the protein level of CCN1 or OCN from the blot were quantified by using Image Lab software and were normalized by the level of GAPDH. The values represent mean ± SD of the fold change of the CCN1 or OCN protein level relative to the TMSCs transfected with the control siRNA from three independent experimental trials. Statistically significant is denoted as *** *p* < 0.001, as determined by Student’s *t*-test.

**Figure 6 cells-11-00534-f006:**
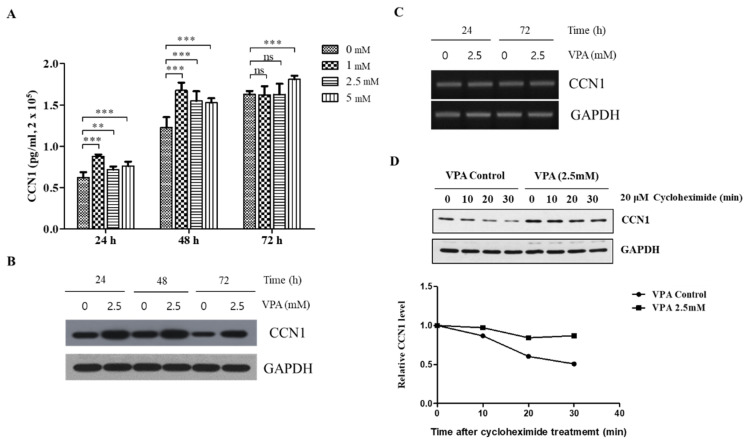
VPA induces the protein level of CCN1 by increasing the stability of CCN1 protein rather than increasing the transcription of *CCN1* mRNA. The TMSCs were seeded at 2.0 × 10^5^ cells/mL of density, followed by VPA exposure at 0, 1, 2.5 and 5 mM for 24, 48 and 72 h. (**A**) The media concentration of CCN1 was measured by using CCN1 ELISA kits. The values represent mean ± SD of the CCN1 concentration (pg/mL) from three independent experimental trials. Statistically significant is denoted as ** *p* < 0.01 or *** *p* < 0.001, which were determined by one-way ANOVA followed by Tukey’s multiple comparisons. (**B**) The protein lysates of the TMSCs exposed to either 0 or 2.5 mM of VPA for 24, 48 and 72 h were collected and was used for Western blotting to measure the level of CCN1 protein, which was presented as a blot. GAPDH was also presented as a reference protein. (**C**) The total RNA of the TMSCs exposed to either 0 or 2.5 mM of VPA for 24, 48 and 72 h were obtained from TriZol method, and PCR was performed to assess the expression of *CCN1*. (**D**) The TMSCs were treated with 20 μM of cycloheximide after 24 h of either 0 or 2.5 mM of VPA exposure. The protein lysates were collected at 0, 10, 20 and 30 min after the cycloheximide treatment to evaluate the time-dependent degradation of CCN1. The protein level of CCN1 was quantified by using Image Lab software and normalized by the protein level of GAPDH. The results are representative of three independent experimental trials, and the values represent mean ± SD of the fold changes relative to the control at 0 min of time points for each respective 0 and 2.5 mM of VPA treatment.

**Figure 7 cells-11-00534-f007:**
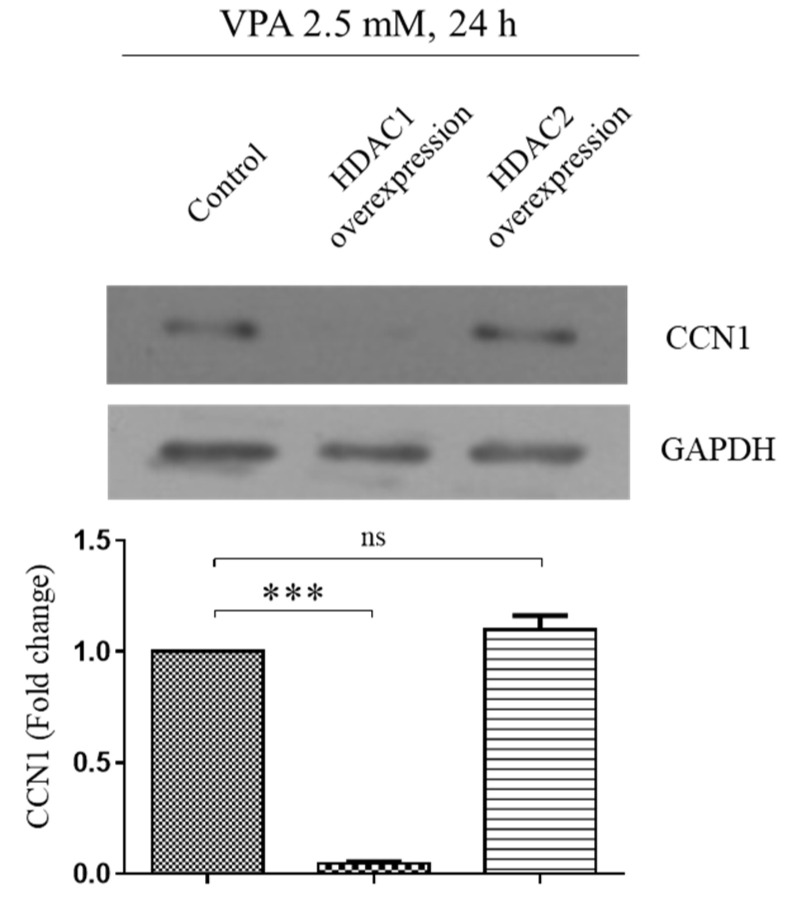
Overexpression of HDAC1 downregulates the CCN1 induced by VPA. We transfected 1.0 × 10^6^ of TMSCs with either 2 μg of HDAC1/pOTB7 or 2 μg of HDAC2/pME18S-FL3 by electroporation, using an Amaxa Human MSC Nucleofector^®^ kit with Amaxa Scientific Nucleofector^TM^ II device, according to the manufacturer’s protocol. The control cells were exposed to the same condition as the transfected group without the inclusion of the plasmid in this study. After the transfection, the cells were treated with 2.5 mM of VPA for 24 h, and the protein level of CCN1 was assessed by using Western blot. The protein level of CCN1 from the blot was quantified by using Image Lab software and normalized by the level of GAPDH. The values represent mean ± SD of the fold change of the CCN1 protein level relative to the TMSCs transfected with the control siRNA from three independent experimental trials. Statistically significant is denoted as *** *p* < 0.001, as determined by Student’s *t*-test.

## Data Availability

The data presented in this study are available upon reasonable request.

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
