# Peer review of "Valproic Acid-Induced CCN1 Promotes Osteogenic Differentiation by Increasing CCN1 Protein Stability through HDAC1 Inhibition in Tonsil-Derived Mesenchymal Stem Cells"

_cells, 2022, doi:10.3390/cells11030534_

Round 1

Reviewer 1 Report

In this study, the authors have shown that valproic acid induced the CCN1 protein expression through inhibition of HDAC1, and enhanced osteogenic differentiation of tonsil-derived mesenchymal stem cells (TMSCs).

Although the study is interesting, it needs more clarifications.

  1. In this study, the authors have isolated mononuclear cells from tonsils, and after 2 weeks, the adherent mononuclear cells were designated as TMSCs. Some monocytes and dendritic cells may attach to the dish. How do the authors eliminate them?
  2. It has been subtitled as “3.4. VPA induces the protein level of CCN1 in a dose-dependent manner but not the protein level of OCN, TAZ, OPN, and RUNX2.”
  3. But in Fig. 3B shows that the protein level of osteocalcin (OCN) is increased.
  4. The authors have not explained when the osteogenic differentiation is induced by VPA, why there is no change in other osteogenic factors such as RUNX2, osteopontin (OPN), etc.
  5. The authors are claiming that “VPA as a specific inhibitor of HDAC1”. While other studies have shown that VPA is not a specific inhibitor, the current claim is big, and the results are not convincing enough to make such a big claim.
  6. What is the use of abbreviations such as VPA and TMSC in the title?    

Author Response

Manuscript ID: cells-1533234
Jan 21st, 2022.

We would like to thank the reviewer for taking the time and effort to provide constructive comments. Please find our revised manuscript (cells-1533234-revised) entitled “Valproic acid (VPA) induced-CCN1 promotes osteogenic differentiation by increasing CCN1 protein stability through HDAC1 inhibition in tonsil-derived mesenchymal stem cells (TMSCs).” as well as the attached pdf file of 'author-coverletter-16752515.v2.pdf' We listed the Reviewer’s comment followed by our response below in the attached pdf file, and all the revised contents were highlighted in yellow in the revised manuscript. We hope that our responses and revisions are satisfactory to the reviewers.

Thank you in advance. I look forward to hearing from you with a favorable response.

Sincerely yours,

Inho Jo, Ph.D. Professor

Department of Molecular Medicine, College of Medicine, Ewha Womans University,

25 Magogdong-ro-2-gil, Gangseo-gu, Seoul 07804 (Republic of Korea)

Tel. +82 2 6986 6267

E-Mail. inhojo@ewha.ac.kr

Reviewer 2 Report

The authors present and intriguing manuscript about the role of Valproic Acid (VPA) as an inducer of osteogenic differentiation of Tonsil-Derived Mesenchymal Stem Cells (TMSCs). They proposed that VPA stabilized the osteogenic factor CCN1 through the inhibition of HDAC1. This manuscript highlights how VPA could influence bone metabolism, nonetheless, to increase the breath of the manuscript and its readability we propose some point to address:

  • Images are all pretty bad quality which makes hard to analyze them. The DPI should be increased.
  • The meaning of CD146 increase should be detailed in the discussion.
  • More explanation of the putative mechanism of CCN1 in osteogenesis would help the reading. For instance, its impact on sclerostin should be mentioned (PMID: 29351359) as it does not affect TAZ, RUNX2 or OPN. Additionally, it is quite surprising that all the osteogenic potential converges through CCN1. Any idea on how CCN1 interact with other regulators of osteogenesis.
  • The mechanism by which HDAC1 inhibition promotes the stabilization of CCN1 should also be elaborated.
  • Line 78 : Which symptoms are alleviated by HDAC inhibitors in cancer and HIV?
  • Figure 3A. How many replicate for these images?
  • There are 4 classes of HDAC encompassing HDAC1 to 11. Only HDAC1 and 2 are discussed. The other HDAC should also be included in the discussion and the conclusion of 3.7 soften as the authors cannot rule out the implication of the other members. Moreover
  • The specificity of VPA for HDAC should be mentioned.
  • More details should be given of about the paradox of VPA that is associated with osteoporosis and the present finding suggesting roles in promoting osteogenesis.
  • For figure 7. Although this data is compelling to let us believe that HDAC1 overexpression is leading to a decrease in CCN1. There is no detail on what the “control” is used. Also, to strengthen this figure one crucial and simple experiment is missing: could VPA reverse CCN1 decrease by HDAC1 overexpression?

Author Response

Manuscript ID: cells-1533234
Jan 21st, 2022.

We would like to thank the reviewer for taking the time and effort to provide constructive comments. Please find our revised manuscript (cells-1533234-revised) entitled “Valproic acid (VPA) induced-CCN1 promotes osteogenic differentiation by increasing CCN1 protein stability through HDAC1 inhibition in tonsil-derived mesenchymal stem cells (TMSCs).” as well as the attached pdf file of 'author-coverletter-17078897.v1.pdf' We listed the Reviewer’s comment followed by our response below in the attached pdf file, and all the revised contents were highlighted in yellow in the revised manuscript. We hope that our responses and revisions are satisfactory to the reviewers.

Thank you in advance. I look forward to hearing from you with a favorable response.

Sincerely yours,

Inho Jo, Ph.D. Professor

Department of Molecular Medicine, College of Medicine, Ewha Womans University,

25 Magogdong-ro-2-gil, Gangseo-gu, Seoul 07804 (Republic of Korea)

Tel. +82 2 6986 6267

E-Mail. inhojo@ewha.ac.kr

Round 2

Reviewer 1 Report

The authors have addressed the criticisms to an agreeable level. 

Author Response

Jan 30th, 2022,

We would like to thank the reviewer for taking time to go over our response; we are glad to hear that the revisions are satisfactory to the reviewers 

Sincerely yours,

Inho Jo, Ph.D. Professor

Department of Molecular Medicine, College of Medicine, Ewha Womans University,

25 Magogdong-ro-2-gil, Gangseo-gu, Seoul 07804 (Republic of Korea)

Tel. +82 2 6986 6267

E-Mail. inhojo@ewha.ac.kr